# Spectral nudging in the Tropics

Breogán Gómez<sup>1,2</sup>, Gonzalo Miguez-Macho<sup>2</sup>

<sup>1</sup>Weather Science, Met Office, EX1 3PB, UK <sup>2</sup>Grupo de Física Non-Lineal, Universidade de Santiago de Compostela, Galicia, Spain

Correspondence to: breogan.gomez@metoffice.gov.uk

Abstract. Spectral nudging allows forcing a selected part of the spectrum of a model's solution with the equivalent part in a reference dataset, such as an analysis, reanalysis or another model. This constrains the evolution in certain scales, typically the synoptic ones, while allowing the others to evolve freely. In a limited area model (LAM) setting, spectral nudging is commonly used to impose the large-scale circulation in the interior of the domain, so that the high resolution features in the LAM's

- simulation are consistent with the global circulation patterns. In a previous study developed over a Mid-Latitude domain, we investigated two parameters of spectral nudging that are often overlooked despite having a significant impact on the model solution. First, the cut-off wave number, which is the parameter determining the scales that are nudged and has a critical impact on the spatial structure of the model solution. Second, the spin-up time, which is the time required to balance the nudging force with the model internal climate and roughly indicates the starting point from when the results of the simulation contain useful
- information. The question remains if our conclusions for Mid-Latitudes are applicable to other areas of the planet. Tropical Latitudes offer an interesting testbed as its atmospheric dynamics has unique characteristics with respect to that further North and yet it is the result of the same underlying physical principles. We study the impact of these two parameters in a domain centred in the Gulf of Mexico, with a particular aim to evaluate their performance related to hurricane modelling. We perform 4-day simulations along 6 monthly periods between 2010 and 2015, testing several spectral nudging configurations. Our results
- indicate that the optimal cut off wavenumber lies between 1000 Km and 1500 Km depending on the studied variable and that the spin-up time required is at least 72 h to 96 h, which is consistent with our previous work. We evaluate our findings in four hurricane cases, allowing for at least 96 h of spin-up time before the system becomes a tropical storm. Results confirm that the experiments with cut-off wavenumbers near the Rossby Radius of Deformation perform best. We also propose a novel approach in which a different cut-off wavenumber is used for each variable. Our tests in the hurricane cases show that the
- latter set up is able to outperform all of the other spectral nudging experiments when compared to observations.

## **1** Introduction

Nudging techniques aim at forcing the evolution of a model towards a reference value that is considered more accurate than the model itself. This is commonly done by adding a relaxation term to the physical tendencies that *nudges* the model's solution with a prescribed strength towards either a set of observations (Bell, 1986; Lyne et al., 1982; Schraff, 1996, 1997) or a gridded

analysis (Davies and Turner, 1977; Stauffer et al., 1991). A variation of the technique, called spectral nudging, allows applying

the forcing on a subset of the spatial frequencies, thus limiting the scales that are affected by the procedure (Miguez-Macho et al., 2004; von Storch et al., 2000; Waldron et al., 1996). Spectral nudging has been used in different applications such as regional climate modelling (Berg et al., 2013; Lucas-Picher et al., 2013; Schubert-Frisius et al., 2016; Shen et al., 2017; Tang et al., 2017), data assimilation (Stauffer et al., 1991; Stauffer and Seaman, 1994), case studies (Silva and Camargo, 2018;

- 5 Wang et al., 2013) or improving turbulent flows (Clark Di Leoni et al., 2018; Yamaguchi et al., 2013). One the first applications of nudging was to improve the representation of tropical cyclones (Anthes, 1974), and spectral nudging has been successfully employed in this field in a number of works (Moon et al., 2018; Tang et al., 2017; Wang et al., 2013). Although it is not strictly required, nearly all nudging studies are performed on limited area model (LAM) applications.
- In spectral nudging, the cut-off wave number is the parameter that defines the scales being forced, allowing the rest to evolve unaffected. Most works choose this parameter by using different criteria, such as the scale they intend to evaluate (Separovic et al., 2012) or the effective resolution of the boundary condition (Liu et al., 2012; Omrani et al., 2013). In Gómez and Miguez-Macho (2017, GM2017 hereafter) we performed a sensitivity study evaluating the impact of different cut-off wave numbers on a LAM domain situated in mid-latitudes where we concluded that nudging the scales above 1000 Km yielded the best results. This is the typical value of the Rossby Radius of Deformation and it suggests that it is best to prescribe only the
- synoptic structure on the LAM and allow it to resolve the finer scales. Other works that evaluated the impact of cut-off wave number in the model simulation, although typically tested fewer values, found similar results (Liu et al., 2012; Schubert-Frisius et al., 2016; Shen et al., 2017; Wang and Kotamarthi, 2013).

In GM2017, we argue that the optimal values of the spin-up time and cut-off wave number can be derived from the synoptic behaviour of the area that is being modelled. Although our conclusions pretend to be generic, the question still stands as to

- whether they can be applied to other areas of the planet. The Tropics have unique characteristics, with large-scale synoptic structures interplaying with strong short-scale unstable phenomena; this is particularly true for hurricanes, as their evolution is the result of the interaction between the synoptic setting and strong local convection. In this work, we aim at finding more evidence to support our hypothesis that the nudging scale should be related to the typical scale of the synoptic systems (i.e. Rossby Radius of Deformation) and not with other factors related to the experimental set-up, such as the resolution of the
- forcing dataset or the model simulation itself. Here we apply a similar methodology as in GM2017 to a LAM located in a Tropical setting to study the impact of cut-off wave number and spin-up time in the solution of the WRF model. First, we study the statistical impact of varying those parameters over a large number of simulations and, second, their effect when simulating particular test cases. We focus our study on the hurricane season, which is particular to the Tropics, and illustrates the interaction between large and small scales in a way not found anywhere else on the Planet.
- This work is organized as follows. In section 2 we describe the experiments. In section 3 we present a statistical analysis of the results and in section 4 we test the validity of our findings in different hurricane cases. Finally, in section 5 we summarize our findings.

5

## 2 Experimental set-up

## 2.1 Model description and configuration

For this work we use version 3.5 of the Advanced Research WRF (ARW) modelling system (Skamarock and Klemp, 2008), set up with a single domain centred over Hispaniola. The domain covers an area from the equator to mid latitudes and has 400x300 horizontal points at 20km resolution and 33 vertical levels (Figure 1). This domain is designed to encompass the entire trajectories of most hurricanes occurring in the area. Model initialisation and forcing is provided by NCEP Global Atmospheric Analysis (GDAS) at 3 hour frequency and the physical configuration of the system is described in Table 1. The WRF model provides grid and spectral nudging capabilities and a full description of the nudging equations can be found in

- section 2 of GM2017.
  In experiments with nudging, this is applied to temperature, humidity and wind. Before WRF version 4, spectral nudging of humidity was not available, so we have implemented it in our version of the model to allow for a comparison of a wider set of variables. We choose to nudge wind in the full atmospheric column, while for temperature and humidity nudging starts to be applied from level 10 to 15 gradually, and then at full strength above this level (see 3.1 on GM2017 for the justification of this
- above the WRF estimation for the planetary boundary layer (PBL), instead of at a fixed level. While our tests suggest that there is not much difference between the two methodologies, the WRF PBL estimation varies substantially horizontally, even across the same synoptic system (i.e. along a cold front). Nudging above a fixed model level ensures that the same nudging force is applied along the domain regardless of the particular synoptic situation. The nudging coefficient, which is effectively a relaxation timescale from the model fields to the reference fields (Stauffer and Seaman, 1990), is set to  $3 \cdot 10^{-4} s^{-1}$ , roughly

approach). This varies slightly from the methodology used in our previous work, where we nudged temperature and humidity

20 corresponding to  $1 h^{-1}$ .

## 2.2 Experiment description

For our experiments, we select 6 periods from years 2010 to 2015, all starting on the 21<sup>st</sup> August and ending on the 21<sup>st</sup> of September of each year. The hurricane seasons from 2010 to 2012 are some of the most active in recorded history, both in named storms and produced damage; whereas the seasons from 2013 to 2015 were below average in terms of hurricane

- 25 occurrence. This ensures that our simulations are representative of a wide variety of situations in the area, which, in conjunction with the extended number of cases considered, gives statistical robustness to our results. Hurricane activity typically peaks in mid-September (Landsea, 1993). For these particular years, starting from the last week of August to the end of September includes a significant number of major hurricanes in the simulations.
- In our experiments we run 12 nudging configurations. Ten of these use spectral nudging with different cut-off values equivalent to spatial scales from 4000km to 125km. Another configuration uses grid nudging and the last one uses no nudging at all, which we refer to as *free run* (Table 2). Each one of the 12 configurations is run for 4 days starting at 00Z every day of each month of the 2010-2015 period, which adds up to 1800 model simulations. Each experiment starting on the same day shares

the same initial and boundary conditions, so that resulting differences are due to the changes in the nudging set-up. The values in the cut-off wave number shown in Table 2 are selected so that they represent approximately the same scale in both horizontal directions. The experiments are tagged with the length scale of the X direction. Free run and grid nudging experiments are given a characteristic nudging length scale corresponding, respectively, to the full domain and twice the model resolution (smallest possible wavelength). Results from sections 3 and 4, as well as findings from our previous work, show that is

acceptable to consider these configurations as asymptotic cases of spectral nudging.

## **3** Results

The evaluation methodology applied to our simulations follows very closely that described in GM2017, in order to make this study and our earlier work in Mid-Latitudes fully comparable. The results from our experiments are contrasted with NCEP

- Global Data Assimilation Analysis (GDAS), which is our driving dataset, to analyse how the model diverges from its boundary condition, and also with ECMWF ERA-interim reanalyses, which represents an independent verification. The fields from both reanalysis datasets are interpolated to the WRF model grid, horizontally and vertically, and we perform the comparisons at different model levels. Scores are calculated for all nudged variables: potential temperature, specific humidity and wind components (which are presented as kinetic energy for convenience). To avoid redundancy in our plots, unless noted, only
- kinetic energy is presented, as results are similar for the other variables. See GM2017, section 4, for a more detailed description of the analysis methodology. Since all of the 1-month periods for integration correspond to the same time of the year, the entire suite of simulations from 2010 to 2015 are included in the calculation to increase statistical robustness.

## 3.1 Power Spectrum and Root Mean Square Distance

The inset panels in Figure 2 depict the power spectrum for each configuration at three different levels. The power spectrum is calculated by first de-trending the fields following Errico (1985) and then applying a FFT to each row (column) of the model. The resulting amplitude coefficients are then averaged for each row (column) and subset for each variable and lead-time. We present results for the X-direction (when applicable) and wind kinetic energy, as results are equivalent for the Y-direction and for the other variables. We show the power spectrum 96 hours after initialisation when all experiments are fully span up (see next paragraph for a justification on this). In general, all experiments show the same behaviour at the larger scales, indicating

- that their synoptic setting contains similar information. As the curves move towards smaller scales, the different nudging configurations start to diverge, revealing the impact of changing the spectral cut-off wave number to shorter wavelengths. The relative difference against GDAS (Figure 2, main panels) details how the spectral structure of the WRF solution changes with respect to its forcing, where grid nudging and free run represent the asymptotic behaviours. On one hand, grid nudging is similar to GDAS at all scales, showing only some divergence with its boundary condition at the smaller scales; on the other,
- the free run shows similar values to GDAS only at larger scales, but it soon separates indicating that is able to generate a more complex small scale behaviour. Spectral nudging cases exhibit an intermediate behaviour, having a similar value to grid

nudging below the cut-off wave number and developing the free run behaviour above it. At higher levels of the troposphere, all experiments are closer to GDAS large scale than at lower levels, reflecting the fact that the atmospheric fields aloft are smoother (i.e. less turbulent and away from the imprint of the terrain). At the 5000 m and 1200 m heights, all experiments show differences with GDAS in the kinetic energy spectra even at the larger scales, indicating that they produce different

- 5 results not only in the short scale but also in the synoptic scale. This is a consequence of experiments with free run and spectral nudging with low cut-off wavenumber diverging from GDAS by developing unrealistic low-pressure systems in the interior of the domain. In section 4 we present a more complete discussion, and in Figure 6 we show an example of this. The Root Mean Square Distance (RMSD) against GDAS and ERA-interim (Figure 3) shows that free run and grid nudging
- have the largest and smallest values, respectively, which is in agreement with the results from the previous paragraph. All experiments start with a low RMSD, reflecting that they start from the same initial condition, and it grows as the lead-time progresses. The time required to achieve a constant distance to GDAS and ERA-interim varies significantly for each experiment and it is a large as 72 to 96 hours for the experiments with the smallest cut-off wave numbers. This indicates that the experiment has reached its internal climate and has fully span up from the initial state. The free run RSMD grows at a steady rate as the lead time advances and it has a similar value for GDAS and ERA-interim, suggesting that the fields are not
- strongly influenced by the boundary conditions. This reflects the fact that the model is developing its own solution in the interior of the domain; which is a consequence of being driven only at the boundaries, with no extra forcing used elsewhere. Applying spectral nudging, even at the smallest wave number, has an immediate constraining effect, preventing the model from separating from its boundary condition. As larger wave numbers are used, the model becomes more similar to GDAS and ERA-interim, and it needs a shorter time to reach a steady behaviour. Ultimately, grid nudging reaches a steady value in
- the shortest time which indicates that this experiment, and spectral nudging experiments with the highest wavenumbers, are very close to GDAS. In fact, the RMSD is small (i.e. almost zero) when compared against this dataset. Here WRF is essentially replicating GDAS field structure and the error against ERA-interim is showing its climatic difference against GDAS. This, and the fact that the error at free run is similar for both datasets, allows speculating that we would find similar results in our statistical analysis across section 3 if we reversed the driving and verification datasets in our experiments.
- The analysis of the Power Spectrum indicates that spectral nudging is very effective at separating nudged and non-nudged scales. The non-nudged scales develop a similar amplitude to those in the free run case, while the nudged scales are-closer to their counterparts in the grid nudging case. Similarly to what it was observed at mid-latitudes, applying large cut-off wavenumbers (i.e. nudging smaller scales) results in the removal of critical small-scale information. Conversely, the experiments with the largest cut-off wavenumber have the lowest RMSD, but this is due to the fact that GDAS and ERA-
- interim have a fairly low resolution and the experiments with less short scale information (i.e. smoother fields) verify better against them. It can be seen that nudging in the largest wavelength represents a substantial improvement in the model solution, due to a better representation of the larger scales (i.e. synoptic situation); however, nudging in the shorter scales makes the model more similar to GDAS and ERA-interim and prevents it from generating finer scale features, which is undesirable.

Therefore, the most appropriate wavenumber should be large enough to prevent WRF from drifting from its boundary condition, but not so high that it does not allow it to develop its internal high resolution dynamics.

## 3.2 Root Mean Square Distance vs Cut-off wave number

Figure 4 (a, c) shows the RMSD against the cut-off wavelength of each experiment for three model levels at 96h lead time,
when most experiments have passed the spin-up phase (see 3.1 for a justification of this). Free run and grid nudging are also included as asymptotic cases, and although they do not have a cut-off wavelength, we use the domain size and twice the model resolution, respectively, to include them in the graph (Table 2). As for Figure 3, results are calculated using all experiments from 2010 to 2015.

Results show that most of the reduction in the RMSD is achieved by nudging the largest scales and the curves flatten out and not much change occurs after a certain inflection point (Figure 4, a, c). Applying nudging above certain wavelength yields almost no change in the RMSD, and according to the results presented in 3.1, it would be at the expense of damping the high spatial frequency contribution from WRF. This is in agreement with what was observed in the mid-latitude experiments and the same conclusion applies here: the cut-off wave number should be selected so that a significant part of the error is reduced, but the high frequency information from the LAM is preserved.

- In GM2017 we used a geometric method to estimate the optimal value of the cut-off wave number which did not consider the physics of the system or the nature of the error. Here we propose an improved methodology that is based on a physical approach. In Jung and Leutbecher (2008) the authors investigated the separate contribution of the planetary, synoptic and sub-synoptic scales to the error of a global ensemble. They estimated that the size of the error was 7 times larger for the synoptic scale than for the mesoscale. In our plots, we indicate (Figure 4, a, c) the point of the curve where the RMSD is reduced to
- 15% of the maximum. The points depicted in the figure are reasonably close to the inflection point, which is quite remarkable as the method does not make any assumption on the shape of the curve. The inflection points for levels between 1000 m and 10,000 m are plotted in Figure 4 (b, d). Results for the RMSD against

GDAS (Figure 4, b) show that kinetic energy and theta inflection points occur around 600 km, slightly lower for specific humidity, and for larger values when they are calculated using the RMSD against ERA-interim (Figure 4, d). This is a

- consequence of RMSD against GDAS drifting to zero as the nudging is extended to more parts of the spectrum, where each experiment shows a slightly lower error that the preceding one (Figure 3). On the contrary, RMSD against ERA-interim shows its greater reduction for the first wavenumbers and quickly reaches its asymptotic value, not changing much afterwards. Therefore, curves for GDAS (Figure 4a) appear straighter than those for ERA-interim (Figure 4c), and its inflection points happen for smaller values than ERA-interim. Since ERA-interim represents an independent verification, the RMSD against
- this dataset is more representative of the model's error and; therefore, we focus our analysis on these results only. Here the inflection points occur for larger values, at clearly separated scales for each variable and with a remarkable consistency in the vertical, which highlights the barotropic nature of the Tropical latitudes.

The average value of the inflection point profiles depicted in Figure 4 (b, d) is of around 2000 km for theta, 1100 km for kinetic energy and 750 km for specific humidity. This result differs substantially from that for Mid-Latitudes in GM2017, where we showed that all variables have similar values of around 1000 km. One possible explanation lies in the fundamental differences between mid-latitude and tropical weather systems. In the former, the spatial structure of the three variables is mostly driven

- by the synoptic setting, where the position of low pressure systems and fronts plays a key role. However, in tropical latitudes, there is a greater variation in scale where, on one hand, temperature is generally smoother and, on the other, humidity varies spatially due to strong and active convection. It is also reasonable to consider that the kinetic energy behaviour would lie somewhere in between, having a generally coarse structure occasionally disrupted by tropical storms crossing the domain. For this reason, WRF gets most of the reduction of the temperature RMSD against ERA-interim for a larger wavenumber than in
- the case of specific humidity. It is easier to reproduce a smoother field such as temperature than it is a more complex one with a high spatial variation like humidity.

One of the main conclusions from GM2017 is that there is a relationship between the most appropriate cut-off wave number and the Rossby Radius of Deformation, which offered a physical interpretation for our results. Nearer to the equator, the Coriolis force weakens, and we expected the optimal cut-off wave number to correspond to a larger scale for a tropical

experimental setting than in mid-latitudes. If we take the kinetic energy as the variable representing the dynamics of the flow, the optimal inflection points for this variable, which is around 1100 km, indicates that, in general, our earlier hypothesis still holds true and offers further support for our results.

## 4 Example cases

## 4.1 Description of hurricanes

- We evaluate our results on 4 hurricanes that occurred between 2010 and 2013 using the same domain (Figure 1), model setup (Table 1), driving dataset (GDAS) and nudging configuration (Table 2) as in the previous section. In addition, we test a new spectral nudging approach (labelled as 3VARS) in which we use different cut-off wave numbers for each nudged variable. This is motivated by our findings from the previous section and we choose them to correspond to 2000 km for temperature, 1100 km for each wind component and 750 km for relative humidity. We added code to WRF to enable us to use this variable
- cut-off wavelength, as it is not available in the standard release. The simulations start between 5 and 6 days before the system evolves into a tropical storm. This spin-up time is a bit longer than the 96 hours that we concluded as appropriate in the previous section, but it should be noted that each of the systems studied here evolved from a tropical depression to a tropical storm and, ultimately, became a hurricane. A longer spin-up time is needed to ensure that the system that will eventually generate the hurricane is appropriately defined in the simulation. To verify our results, we use the centre position, pressure and maximum
- wind from the National Hurricane Center reports (Table 3). Each hurricane's observed tracks can be seen in Figure 5 tagged with the OBS label. The hurricane cases have been selected to represent different trajectory paths representative of typical

trajectories in the zone. We briefly describe them here; a comprehensive description of each one of the systems can be found in the NHC reports.

- Earl originated in the Tropical Atlantic. It travelled westward, turning north in the Caribbean and continued moving parallel to the US east coast until it dissipated east of Labrador.
- Isaac followed a similar path to Earl in its early stages, but continued traveling westwards, crossing the Gulf of Mexico until it dissipated over the southern US.
  - Michael originated in the Atlantic at a relatively high latitude and had a wiggling track that stayed over the Atlantic Ocean throughout its lifespan.
  - Ingrid was a short lived hurricane with a trajectory that started and ended in the Gulf of Mexico.

# 10 4.2 Results

5

Figure 5 depicts the hurricane tracks simulated by each one of the experiments for the four hurricane cases alongside their observed positions. In all cases, nudging for nearly any configuration is able to maintain the hurricane close to the observed path while the free run experiment does not prevent any of the hurricanes from drifting to unrealistic tracks. A similar deviation occurs in experiment SP 4000 for Michael and Ingrid, which can be explained by the fact that their trajectories develop on a

15 smaller area and applying nudging in the very long scales is not sufficient to prevent them from evolving unrealistically. The tracks for Earl and Isaac, which both travelled over a larger area than either Michael or Ingrid, are accurately modelled with all nudging set ups.

Figure 6 shows maps of mean sea level pressure contours every 4 mb, 24 h accumulated precipitation and centre position for the 8 experiments. Fields from ERA5 (MSLP), TRMM (precipitation) and the centre position from NHC reports are also

- included as verification. Plots represent the fields 5 days after the system turned into a tropical storm and 10 days from the start of the simulation. We show results only for Isaac as comparable results were found for the other hurricane cases. The benefits of using an appropriate nudging set-up extends to all parts of the model domain, and not only to a particular area, as it can be seen in the experiments with smaller cut-off wave number (FR 8000, SP 4000 and SP 2000), which develop more
- fine scale features than other experiments. In this case, it has a detrimental effect on the solution as many mesoscale cyclones that are not present in the observations are simulated by the model. For nudging scales between SP 1500 and SP 1000, the synoptic fields resemble the general structure of the ERA5 and TRMM fields with no presence of the bogus systems observed in the experiments with smaller cut-off wave numbers. This suggests that nudging only the larger parts of the spectrum is sufficient to ensure the general synoptic structure is correctly placed. The 3VARS experiment, although having a fairly large nudging scale for temperature (whereas not so large for wind and humidity) is able to produce a similar synoptic structure as
- in the observations, with no trace of the spurious cyclones.

To have a better understanding of how each experiment is performing, in Figure 7 and Figure 8 we show the accumulated error for hurricane centre position (top row, labelled AEPOS), centre pressure (middle row, labelled AEPRS) and maximum wind (bottom row, labelled AEWND) for Earl and Isaac and Michael and Ingrid respectively. The experiments that do not represent

the tracks appropriately (i.e. FR in all cases, and SP 4000 in Ingrid and Michael) are omitted in the plots as their estimation of the centre pressure and max wind is irrelevant in this context.

The AEPOS shows that, the more parts of the spectrum that are nudged, the better the centre is positioned in the simulation, and that GR 40 and SP 500 consistently perform best in all cases. The two experiments benefit from using the GDAS data

- assimilation component, which result in these two cases having an accurately positioned centre. However, this is not the case for AEPRS and AEWND, as results are more inconsistent for these two experimental settings. In general, experiments with cut off wavenumbers between 2000 km and 1000 km are the best performers for central pressure and wind. Without considering the 3VARS experiment, for Ingrid and Michael the best result occurs in AEPRS for SP1500 and in AEWND for SP1300, and for Isaac and Earl there is not a clear best configuration. Since the optimal cut-off wavenumber depends on the area, and Ingrid
- and Michael have a trajectory that is constrained into a relatively small one, a particular cut-off wave number setting is able to nudge the optimal large scale features allowing the LAM to develop the local features of the area. It is also noteworthy that Ingrid, which is located further south where the Coriolis force is weaker, performs better with a larger cut-off wave number than Michael. On the other hand, for Isaac and Earl there is not a clear best configuration, reflecting the fact that both systems have a long trajectory that is not localised over a small area. Therefore, there is not a single optimal cut-off wavenumber, as
- the synoptic structures that interact with the hurricane change in scale as the system moves northwards. Considering the three scores at the same time, the 3VARS experiment is consistently among the best performing experiments in all variables and hurricane cases, showing the benefits of tuning the cut-off wave number for each variable.

The accumulated errors at the end of the evaluation period normalized by the 3VARS case are presented in Figure 9. Variables will appear smaller or larger than 1 depending on whether they have a smaller or bigger error, respectively, than the 3VARS

- case. Similar to the previous graphs, since FR 8000 and SP 4000 are not able to reproduce the hurricane track, are not presented here. The total height of the column in Figure 9 accounts for the accumulated skill performance across all variables and hurricane cases. This figure summarizes whether a particular case is performing better or worse than 3VARS. Results confirm that the 3VARS setup outperforms the other cases overall, not being necessarily the best at each particular score, but giving a consistently good performance across all the three considered variables. The only exception is SP 500, which shows a similar
- overall performance to 3VARS, but this is largely due to its better accuracy representing the centre position of the hurricane (darkest colours), while having poorer performance in simulating both the centre pressure and maximum winds. This highlights the fact that this experiment, together with GR 40, is able to give a good centre position estimation because it is strongly nudged towards the GDAS fields, but it fails to develop a hurricane system well enough to deepen the centre pressure and, consequently, to have sufficiently strong winds. On the other hand, the 3VARS experiment offers a more balanced prediction
- of all variables where the centre position estimation is still good compared with most of the experiments and gives a good estimation of the maximum winds and centre pressure.

#### **5** Summary and conclusions

The work presented in this paper is strongly connected with the ideas and concepts introduced in Gómez and Miguez-Macho (2017) where we studied the impact of the cut off wave numbers and spin up time in a model simulation using spectral nudging. Our conclusions suggested that the parameter selection is related with the synoptic characteristics of the area and, since the

- domain was located in a mid-latitude setting, the question remained if the same conclusions were valid at other latitudes. In this work, we test our ideas in a tropical setting during the hurricane season.
  - For this study, we ran the WRF model over a large domain around the Gulf of Mexico including a substantial portion of the North Atlantic Ocean, where most tropical storms that ultimately become hurricanes are generated. Our modelling set-up is run for 96 h starting every day between late August and late September in the 6 years between 2010 and 2015. This combines
- the 2010-2012 period, one of the most active hurricane periods in history, with the 2012-2015 period, which was below average, thus, ensuring a wide variety of cases was included in the study. For each day, 12 different experiments were run, including 10 different spectral nudging configurations with varying cut off wave numbers from the lowest to the highest, a grid nudging and a free run setting.

The temporal evolution of the RMSD indicates that the spin-up time during which the model balances with the nudging effect

- is between 72 h and 96 h, which doubles the value found in the mid-latitudes. We speculate that this is linked to the model initial conditions, which, with the exception of water vapour, do not contain any other water species and are set to zero at initialisation. The process of developing clouds liberates latent heat and creates a disruption at the early stages of the simulation. Tropics are typically more covered with clouds than Mid-Latitudes (International Satellite Cloud Climatology Project, n.d.) and, consequently, WRF model needs to develop a larger amount of cloud water from the initialisation closer to the equator
- than in the poles, thus creating a larger disruption in this area. The same reasoning can be applied to the other water species (i.e. ice, snow, hail, etc.) explaining why it takes longer to reach a balance with the nudging force in the Tropics than in Mid-Latitudes.

Our analysis of the spectral structure of the model solution reveals that spectral nudging is capable of separating the nudged scales from those evolving freely, making the model solution similar to the reference fields below the cut-off wavenumber and

- allowing the model to develop its own dynamics above it. Nudging clearly improves the predictability of the system and applying it at the larger scales results in the greatest reduction in the error. As nudging is extended to larger parts of the spectrum, little benefit is obtained, and it happens at the expense of dampening the higher spatial frequency phenomena in the simulation. This was particularly notable in the hurricane cases presented, where the simulations with the larger cut-off wave numbers underestimated the hurricanes' centre pressure and maximum wind speeds.
- Results suggest that the cut-off wave number should be selected so that it provides a significant reduction in the error without dampening the finer scale detail, which is the relevant contribution of the LAM. Our findings suggest that this optimal cut-off wavenumber occurs at different scales for each nudged variable, being 2000 km for temperature, 1100 km for wind and 700 km for humidity. These values are in agreement with each variable's individual synoptic characteristics, accounting for the

5