# Peer review of "Spectral nudging in the Tropics"

_Earth System Dynamics, 2020_

## Referee Comment (RC1) · Anonymous Referee #1 · 14 Dec 2020

Review of article "Spectral Nudging in the Tropics" by Gomez and Miguez-Macho.

This paper examines the impact of spectral nudging in the tropical-subtropical area centered in the Gulf of Mexico and tropical cyclone simulations using WRF model. The authors find that 72-96h time is necessary for model to fully spin-up, and cut off wavenumber in between 1000-1500km is optimal to perform best. The authors also investigated the role of spectral nudging in hurricane case simulations. Overall, I find this is an informative study and helps readers for better understanding the role of spectral nudging in the numerical simulation. However, some of the arguments in this paper sound too generic and the results are questionable due to the lack of physical explanation. Therefore I am recommending major revisions to this manuscript before I can consider it is suitable for publication. My comments sorted by major and specific are

below:

Major comments: 1) This manuscript is entitled "Spectral Nudging in the Tropics", but the model domain is centered in the Gulf of Mexico (domain covers roughly 0-45N?), where strictly speaking, it is not tropical area. The reviewer wants to know if the similar results still hold in the more conventional tropical area like 30S-30N where tropical wave activities are more active? 2) Since authors use GDAS as the nudging input, using ERA-interim as an independent verification is a good choice. However, I think authors can make more comparisons with some satellite/in-situ observation, which does not or less use data assimilation. For example, authors can reproduce Figure 3 with the usage of TRMM/NCEP Stage IV precipitation. 3) I agree that spectral nudging with high cut-off wavenumber make tropical cyclone (TC) simulation very analogous to the ERA5 results. However, since both ERA5 and GDAS have already assimilated TC circulations (https://journals.ametsoc.org/view/journals/wefo/26/6/waf-d-11-00045_1.xml?tab_body=fulltext-display), it's not surprising to see these results. However, if model simulation is extremely similar to the ERA5 or other reanalysis with nudging, what is the benefit of running WRF model? Comparing free run and nudging runs, it clearly to see that, with stronger nudging, the more model internal dynamics/physics lose.

Specific comments: 1) WRF uses terrain-following sigma coordinate. The same sigma level does not share the same height level, then why authors claim sigma levels 11, 19 and 23 are at 1200, 5000 and 10400m high? Why not interpolate to the pressure/z level directly? 2) Some of the figures are in poor quality and without useful captions. For example, Figure3 needs an individual legend even though it shares the same one with Figure3. This really bothers me to figure out the line types and corresponding experiments. 3) Figure 6 is really hard to read clearly. Although ERA5 is the state-of-the-art reanalysis, it is still not the observation, especially for the hurricane case studies. What is the suggested TC MSLP at this time by NHC? What is the value from the ERA5? 4) Again regarding Figure 6, based on the eyeballing, it seems that

although the free run (FR8000) has the very departed TC center location comparing to the observation, it has the strongest intensity in terms of the MSLP, as the isobars are heavily overlapped. For high wavenumber spectral nudging and analysis nudging runs, on the other hand, the TC center is nicely reproduced but the TC intensity is quite weaker than the free run. Is that because high wavenumber nudging constrain too much to the model internal TC dynamics? 5) In Figure 7 and 8, what does the dashed vertical line stand for? What do the "X:4;Y:3" like ratios mean? I may miss something in the manuscript main text, but it will be much more friendly for the readers to better understand the plots if authors can add more useful information in the Figure captions. 6) Figure 9 is also hard to understand. Since all errors are normalized by the 3VARS, does each small box in 3VARS column stands for the unified error (as 1)? This manuscript really needs a better figure captions. 7) Page 8, Line13-17: This is too generic. Is that really because of the short tracks of Michael (2012) and Ingrid (2013)? Not due to the relative weak steering flow and strong TC internal dynamics? 8) Page 11, Line 8-11: "However, the best prediction of the hurricane intensity" I really didn't see any analysis regarding to the prediction in this paper. Authors made suites of TC hindcast simulations, and find that 3VARS gives the most analogous results comparing to the NHC observations. However, authors use GDAS as the nudging inputs. In the true prediction, you cannot nudge anything that has not happened! If authors really want to disentangle the prediction problem, authors need let model run freely in forecast mode.

---

## Referee Comment (RC2) · Anonymous Referee #2 · 21 Dec 2020

In this paper, the authors investigate the impact of spectral nudging on the WRF simulation a domain that covers the Gulf of Mexico. For the first part of this paper, the authors summarize the behaviors of different spectral nudging settings over this domain with ERA-Inteirm as reference. The interesting part is on the second half of this paper: the authors proposed to apply different nudging scales to different model variables and tested its impact on simulation of tropical cyclones. The results show that compared with a uniform spectral nudging for all variables, a variable-dependent nudging leads to smaller intensity errors (i.e., maximum wind, center SLP).

While this paper shows detailed behaviors of spectral nudging, the discussion of Figure 2 & 3 seems not solid, and it needs to be revised. Due to unknown reasons, Figures 1-6 seem of low-resolution, which makes interpretation difficult. Labels and legends of

some figures are also missing. Considering all these, I recommend a major revision for this paper.

Major comments:

1. Page 2, Line 20: The authors say "In this work, we aim at finding more evidence to support our hypothesis that the nudging scale should be related to the typical scale of the synoptic systems and not with other factors related to the experimental set-up, such as the resolution of the forcing dataset or the model simulation itself. ": Why do authors stress SYNOPTIC system here? Should it be large-scale motion of your simulation? For example, for different applications, its large-scale motion might refer to different wavenumbers, but nudging is preferred to constrain the large-scale motion, instead of small-scale one. The boundary to separate these two scales is based on researchers' own needs. In addition, what do you mean exactly for "model simulation itself"? Can you be more specific?

2. Though the spin-up time is not the focus of this paper, but from authors' Figure 3, it seems spin-up time is not sensitive to the configuration of spectral nudging: the RMSD of most experiments get saturated after 10 hours for WRF-GDAS case, while longer for WRF-ERA interim. If this spin-up time is almost similar for all configurations, what's the reason to inspect this?

3. It seems the focus (or the innovative part) of this study is about the influence of variable-dependent spectral nudging over the simulation of tropical cyclones. It might be good to change the current title to something like "variable-dependent spectral nudging for the simulation of tropical cyclones". The original title is misleading.

4. The legend of 3 is missing. Though the authors claim Figure 2 & 3 share the same legend, the line types in these two figures are different. Please revise these figures.

5. If I match the color correctly, the discussion of Figure 3 seems not solid: (1) Page 5, Line 15 "This reflects the fact that the model is developing its own solution... Applying
spectral nudging, even at the smallest wave number, has an immediate constraining effect, preventing the model from separating from its boundary condition.": How did you get the conclusion that nudging prevents the model separate from BOUNDARY CONDITION? In spectral nudging, you are regulating the large-scale motion of WRF simulation with the GDAS large-scale motion. This alone, in my view, is enough to enable WRF to have similar large-scale motion as GDAS. Boundary condition doesn't need to be involved in this process. (2) Page 5, Line 25 "The analysis of the Power Spectrum indicates that spectral nudging is very effective at separating nudged and non-nudged scales. The non-nudged scales develop a similar amplitude to those in the free run case, while the nudged scales are-closer to their counterparts in the grid nudging case. ": This argument seems not solid. From figure 2, we know the kinetic energy spectrum (which is a statistical term) for large-scale is similar, but their specific patterns (e.g., figures of streamline, and geopential height) can be different. It seems a more straight-forward way to support your argument is to plot the 2D maps such as streamlines for different scales. From these figures, we can directly check if the largescale pattern for the spectral nudging case is similar to those of GDAS or ERA-Interim, while their small-scale features are different. (3) Page 4, Line 31 "It can be seen the nudging in the largest wavelength represents a substantial improvement....": Why it's a substantial improvement? In Figure 2 & 3, you calculate the RMSD of your WRF simulation from GDAS and ERA-Interim. The larger RMSD here only indicates larger deviation from GDAS and ERA-Interim, which is expected.

6. For all the map figures, the latitude and longitude are not shown. Please add them.

Detailed comments: 1. Please add a legend for figure 3. The line types shown in this figure are not consistent with those as in Figure 2.

2. What are those markers in Figure 4(a) and (c)? Are those inflection points or the points where the errors are reduced to a 15% of their maximum value? If they are inflection points, why they are not on the line for QVAPOR in 4(a)?
3. Figure 6: The color chosen for the precipitation field are difficult to interpret, and the labels of MSLP contours are missing. Please reconstruct this figure.

---

## Referee Comment (RC3) · Anonymous Referee #3 · 22 Dec 2020

This manuscript presents an interesting contribution in the area of limited area modelling. It discusses the optimal spectral nudging setup for downscaling reanalyses over domains centred in the tropical regions. The manuscript also discusses the impact of nudging configurations on the simulation of tropical storms. It thus provides important technical information for configuring the downscaling experiments in the tropics. The paper is overall very well written and is a pleasant read but, in my opinion, some parts still need to be improved before the paper is ready for publication. I recommend a major revision.

Major comments:

1. Please specify the units of kinetic energy spectra, where appropriate (e.g., Fig. 2). Are the spectra calculated from wind components as (S(u)+S(v))/2 or from kinetic

energy field as S((uˆ2+vˆ2)/2)?

2. The method to estimate the optimal value of the cut-off wave number from inflection points in Fig 4, seems to me to be presented in an upside-down manner. In GM2017, you used a geometric method to obtain the wave number that corresponds to the inflection point of the RMSD curve, whereas in the present manuscript you rely on considerations in Jung and Leutbecher (2008), where it was found that the size of error was 7 times larger for the synoptic scales than for the mesoscales. Based on this finding, the present manuscript proposes a definition of the wavelength delimiting the synoptic scales and mesoscales as the one where the RMSD is reduced to 15 % (∼1/7) of the maximum RMSD value. Such a breakdown seems to me arbitrary and unconvincing. Even Jung and Leutbecher (2008) clearly state that "the actual choice of the wavenumbers used for the breakdown is somewhat arbitrary given that there is no clear gap in the spectrum of meso-scale to planetary-scale atmospheric motions in the extratropics." It seems more natural to me to present the findings in the opposite way: use the geometric method (as in GM2017) to find the inflection point and then quantify how close it is to the point where the RMSD is reduced to 15 %. This would be a more appropriate approach to try to understand the physical meaning of the inflection point and the corresponding change of RMSD slope. The paper could also benefit if more details are provided about the comparison of the two methods.

3. I found the systematic presence of inflection points at different levels and variables in Fig. 4 quite interesting. There may be multiple explanations for this phenomenon. First, spectral slopes flatten between the synoptic and mesoscales (e.g., steep -3 towards a less steep -5/3 slope of kinetic energy spectrum). Could the slope change in RMSD be explained by the slope change in the field itself? Second, in experiments where the cut-off wave number is small, only the scales well resolved by the reanalyses are nudged and RMSD dependence on the cut-off wavenumber is likely governed by error reduction via improving the shapes and timing of large-scale features. On the other hand, when the cut-off wave number is large, beyond the effective resolution of the

reanalyses, RMSD slope in the range of high cut-off wavenumbers could be governed merely by the degree of smoothing of fine scales induced by the spectral nudging, which, I guess would result in a less steep slope. It would be nice to hear your thoughts on these ideas.

4. Have you experimented with the nudging e-folding time? Does the optimal cut-off wave number for the position, centre pressure and max wind speed depend on the e-folding time?

5. Does this study have any relevance for downscaling GCM simulations of the present or future climate? The conclusion section could benefit from a short discussion on this topic.

Specific comments:

1. Please specify the units of kinetic energy spectra, where appropriate (e.g., Fig. 2). Are the spectra calculated from wind components as (S(u)+S(v))/2 or from kinetic energy field as S((uˆ2+vˆ2)/2)?

2. Pg.5, L12: "a large as" -> "as large as".

3. Pg.5, L16: "the domain, which is" -> "the domain, which is" or "the domain; this is..."

4. Pg.5, L25: "Power Spectrum" (no need for capital letters).

5. Pg.5, L26-27: "The non-nudged scales develop a similar amplitude to those in the free run case, while the nudged scales are-closer to their counterparts in the grid nudging case." This seems to be true only in the upper air (level 23). In the lower troposphere there is a flat amplitude increment affecting all scales in all experiments SP>=2000. I guess this is due to the bogus cyclones that develop when the cut-off wavenumber is too low.

6. Pg.7, L9: "WRF gets most of the reduction of the temperature RMSD against ERA-interim for a larger wavenumber". Did you mean larger wavelength? There seems to

be a similar confusion in the caption and annotations of Fig. 4.

7. Pg.8, L27: "nudging only the larger parts of the spectrum". Do you mean nudging only the larger wavelengths?

8. Pg.11, L6: "intensity ," please remove the blank.

9. Fig. 2: Should the meaning of line types and colouring be given in the figure caption?

10. Fig. 2: What is 'db' in the caption of Fig. 2?

11. Fig. 3: Not clear which row corresponds to which level. Please annotate the panels properly. Also, I think it would be more elegant to have a unique y-axis range for comparisons against GDAS and ERA-Int.

12. Fig. 3: Is the unit here really Joule? I think we are talking about energy per unit mass or unit volume, right?

13. Fig. 3: A small inconsistency – the authors state the colours and symbols are the same as in Fig. 2 but here we have dashed-dotted and dotted lines for SP750-4000 vs. dashed and dotted in Fig. 2.

14. Fig. 4: The quality of this figure should be improved. Annotations are too small and blurred (almost unreadable, at least for me). In the figure you printed "Wn" – does it stand for wave number or wavelength? If the latter, please add units.

15. Fig. 4: Please consider revising the figure caption. "Panels a & c show results theta". Why not "Panels a & c show results for potential temperature"? Also, what are "WL numbers" and how are they related to Wn in the figure? Such inconsistencies make the read a bit difficult. Also, levels printed are 10,18,22 and not 11,19,23. Vertical staggering?